# Determinants of severe maternal outcome in Keren hospital, Eritrea: An unmatched case-control study

**Henos Kiflom Zewde** ⓘ *

Departement of Family and Community Health, Ministry of Health Anseba Province, Keren, Anseba, Eritrea

* heniutd@gmail.com

## Abstract

### Background

In the past few decades, several studies on the determinants and risk factors of severe maternal outcome (SMO) have been conducted in various developing countries. Even though the rate of maternal mortality in Eritrea is among the highest in the world, little is known regarding the determinants of SMO in the country. Thus, the aim of this study was to identify determinants of SMO among women admitted to Keren Provincial Referral Hospital.

### Methods

A facility based unmatched case-control study was conducted in Keren Hospital. Women who encountered SMO event from January 2018 to December 2020 were identified retrospectively from medical records using the sub-Saharan Africa maternal near miss (MNM) data abstraction tool. For each case of SMO, two women with obstetric complication who failed to meet the sub-Saharan MNM criteria were included as controls. Bivariate and multivariate logistic regression analyses were employed using SPSS version-22 to identify factors associated with SMO.

### Results

In this study, 701 cases of SMO and 1,402 controls were included. The following factors were independently associated with SMO: not attending ANC follow up (AOR: 4.53; CI: 3.15–6.53), caesarean section in the current pregnancy (AOR: 3.75; CI: 2.69–5.24), referral from lower level facilities (AOR: 11.8; CI: 9.1–15.32), residing more than 30 kilometers away from the hospital (AOR: 2.97; CI: 2.29–3.85), history of anemia (AOR: 2.36; CI: 1.83–3.03), and previous caesarean section (AOR: 3.49; CI: 2.17–5.62).

### Conclusion

In this study, lack of ANC follow up, caesarean section in the current pregnancy, referral from lower facilities, distance from nearest health facility, history of anaemia and previous caesarean section were associated with SMO. Thus, improved transportation facilities,

**Data Availability Statement:** The complete data set supporting the conclusion of this manuscript have been provided as a supporting information in this manuscript.

**Funding:** The author(s) received no specific funding for this work.

**Competing interests:** The authors have declared that no competing interests exist.

**Abbreviations:** MNM, maternal near miss; CEmOC, comprehensive emergency obstetric care; WHO, world health organization; SMO, severe maternal outcome; ANC, antenatal care; SPSS, statistical package for social sciences; OR, odds ratio; CI, confidence interval.

robust referral protocol and equitable distribution of emergency facilities can play vital role in reducing SMO in the hospital.

## Introduction

An important target of Sustainable Development Goals (SDGs), which urges the international community to reduce maternal mortality rate (MMR) to less than 70 per 100,000 live births by 2030, [1] is still far from reach. Despite huge international efforts to reduce its incidence, maternal mortality remains one of the most important public health problems. Even though the global maternal mortality rate fell from 442 to 211 per 100,000 live births within the past twenty years [2], the number of maternal deaths is still high in most developing countries. Sub-Saharan Africa countries bear the highest burden of maternal mortality contributing to 66% of the total maternal deaths reported globally [2].

The high burden of adverse maternal events in some resource-poor areas of the world is a clear manifestation of the inequalities in health service access, and highlights the gap between the rich and poor. Four largely preventable obstetric complications (bleeding, infections, pre-eclampsia/eclampsia, and unsafe abortion) account for 80% of all maternal deaths [3]. Most maternal deaths are avoidable since there are well-established healthcare solutions to prevent and manage obstetric complications [4]. Yet, majority of maternal deaths that occur in developing countries are largely preventable [5,6].

Information on the determinants of MNM and mortality is crucial for general assessment of resource requirements, appropriate policy formulation, as well as prioritization and effective implementation of interventions. Previously a number of distant (socio-economic and demographic characteristics of women) and intermediate (obstetric history, preexisting medical conditions, and reproductive health characteristics) factors were revealed as determinants of maternal mortality and morbidity [7–10].

The WHO MNM tool [11] recommends investigation of all cases of MNM and maternal death to gain better understanding of the circumstances surrounding maternal death and severe maternal morbidity. This principle is based on the assumption that all maternal deaths experience at least one life threatening condition (organ dysfunction), and therefore are similar to cases of MNM in every aspect except for the ultimate outcome of interest. Even though there are several studies in the literature that focus on the determinants of either maternal mortality [12,13] or MNM, [7,8,14] there is still scarcity of research focusing on potential factors associated with severe maternal outcome (SMO).

Eritrea is one of the sub-Saharan countries with high rate of maternal mortality. Despite this fact, studies on the subject of maternal morbidity and mortality are virtually inexistent in the country. Hence, there is lack of knowledge regarding factors associated with SMO in Eritrea. The aim of this study was, therefore, to assess determinants of SMO among mothers admitted to Keren Regional Referral Hospital.

## Methods

### Study design, setting and source population

To address the objectives of this study, unmatched case-control study was conducted in Keren regional referral hospital. Located in the town of Keren, this hospital is the only reference hospital for high complexity obstetric care in the entire Anseba region. The hospital serves both self-referred women and those referred from community hospitals, health centers, and health

stations across all subzones of Anseba Region. It provides comprehensive emergency obstetric and newborn care (CEmOC) services. Approximately 549,000 people who reside within its immediate catchment area are served in the hospital. All women who were in labor, delivered or aborted, or within 42 days of postpartum and admitted in maternity or emergency wards of this hospital from January 2018 to December 2020 were the source population for this study.

### Inclusion criteria

**Cases.** All women admitted to Keren Hospital during the study period for the treatment of pregnancy related complications, having delivered, or within 42 days of termination of pregnancy, and who fulfilled at least one of the conditions stated in the MNM tool adapted for sub-Saharan Africa [15] were enrolled as cases in the current study. Similarly, women who died while pregnant, during childbirth, or within 42 days of termination of pregnancy, irrespective of the duration and the site of the pregnancy, from any cause related to or aggravated by the pregnancy or its management but not from accidental or incidental causes were also included as cases.

**Controls.** Women admitted to the hospital with obstetric complications who failed to satisfy the sub-Saharan MNM criteria were enrolled as controls for the present study. For each case of SMO, two controls who were admitted in the same day were randomly selected.

### Exclusion criteria

Women whose medical records were unavailable at the time of data collection were excluded from the study. Moreover, any woman whose medical record contains missing information on two or more variables was left out from the study.

### Data collection process and materials

First, the data collectors identified all cases of MNM and maternal death that occurred during the entire study period. They used the sub-Saharan MNM data abstraction tool to identify women who experienced MNM event. Then, information regarding the potential proximate and distant determinants of SMO was collected from all cases of SMO and their respective controls using a structured data abstraction tool. The tool was prepared after thorough review of literature. Data were collected from the medical records and registries of antenatal care ward, delivery ward, obstetric ward and emergency ward. For every woman, data related to socio-demographic characteristics, obstetric history, obstetric conditions, history of comorbidities, and underlying complications were collected. Prior to the commencement of the actual study, pretest was conducted in the hospital using 10% of the sample size to ensure the appropriateness of the tool. Moreover, each data collector completed the data abstraction tool for all women who were selected in the pretest to ensure that there is substantial agreement amongst them. The Cohen's Kappa Statistic was then used to evaluate the inter-observer agreement, and its value was above 0.9 indicating a near perfect concordance between the data collectors.

### Sample size determination

The sample size for this study was calculated using Epi info-7 software employing a method for unmatched case-control studies. While calculating the sample size the following assumptions were made: confidence level of 97%, power of 80%, case to control ratio of 1:2, proportion of controls exposed 1.5%, and proportion of cases exposed 15.3%. The proportion of exposure among cases and controls was taken from an Ethiopian study [7]. After careful consideration of all exposure variables included in the study, the one yielding the largest sample

size (no ANC follow up) was considered in the sample size calculation for the current study. Based on the above assumptions, a minimum sample size of 188 cases and 376 controls was required. After adding 10% for data unavailability, the final sample size was calculated to be 207 cases and 414 controls. However, we opted to include all SMO cases identified during the entire study period along with their controls in order to increase the power of the study.

## Data analysis

The collected data was coded, entered and cleaned using Epi info-7 software program and analyzed using SPSS Version-22 computer software. SMO was the dependent variable for the present study. The following variables, on the other hand, were the independent variables: age, parity, antenatal care attendance, birth interval, gestational age, mode of delivery, mode of admission, proximity of residence to hospital, preexisting chronic medical conditions, history of anemia, history of caesarean section, and history of abortion. Descriptive statistics were used to summarize baseline characteristics of cases and controls. Potential differences between cases and controls with regard to the distribution of independent variables were assessed using Pearson's Chi square test ($\chi^2$). Binary logistic regression analysis was also employed to determine the influence of each independent variable on SMO individually. Multivariate analysis was then employed to find out whether the factors that were found to be significant in the bivariate analysis remain independently associated with SMO. Only variables that showed statistical significance at $p < 0.25$ were considered in the multivariate logistic regression model. For each variable the crude and adjusted odds ratio, confidence interval, and p-value were reported. The level of significance for the bivariate and multivariate logistic regression analysis was set at 0.05.

Fitness of the final model was assessed using the Hosmer-Lemeshow (HL) test. This test suggested that the model was best fit for the data as the HL Statistic was found to be insignificant ($>0.05$). Values of Tolerance and Variance Inflation factor (VIF) were also checked to ascertain the absence of multicollinearity between the independent variables involved in this study.

## Data quality assurance

Data for the current study was collected by three competent nurse midwives who have been providing CEmOC services for more than five years in the hospital. They were given training on how to identify relevant registers, how to extract information from them, and how to handle unclear and missing data. Clear and detailed definition of every criterion in the SSA MNM tool [15] was given to all data collectors to minimize errors in identifying cases of MNM. Likewise, the International Classification of Diseases (ICD-10) definition of maternal death [16] was used to include eligible cases of maternal death. Data collectors were strictly informed to contact the principal researcher in case of any ambiguity or uncertainty; such ambiguities were then resolved based on consensus reached between the researcher and the data collectors. The principal researcher closely monitored the entire data collection process, and registers were revisited in case of any doubt.

## Ethical consideration

The entire study process was strictly abided by acceptable ethical standards. Approval was sought and granted from the Provincial and National Health Research Review and Ethics Committees. Adequate explanation was given to all concerned bodies regarding the purpose and benefits of the study. The study was conducted in accordance with the guidelines for research outlined by the National Research and Ethics Committee and the Helsinki declaration

of 1975, as revised in 1983. Written consent to review medical records was obtained from the medical director of Keren Hospital. The need to obtain informed consent was waived by the Research and Ethics Committee as this is the standard procedure for research activities entirely based on routinely collected data. In order to revise records in case of doubt and to avoid repetition of cases recorded in multiple registers in different wards, essential participants' identifiers were recorded in a logbook. The logbook was kept in a secure cabinet until data collection period was over, and all recorded information was destroyed soon after the completion of the study. Likewise, all data obtained using the data abstraction tool were de-identified following completion of the study, so that none of the collected information could be tracked back to any individual woman.

## Results

During the 3-year period reviewed, 701 (SMO) cases were identified (662 cases of MNM and 39 maternal deaths). We also included 1,402 women who delivered without any notable complications during the same period to be used as controls in this study.

### Socio-demographic, obstetric, prenatal and perinatal characteristics of study participants

There was no significant difference between mean age of SMO cases (27.6) and controls (28.2). Larger proportion of women with SMO did not attend antenatal care services compared to controls (p<0.001). Moreover, compared to the control group, cases of SMO tended to have higher parity, have premature delivery, experience abortion, be referred from lower facilities, and reside 30 kilometers or further from the hospital, all of which were statistically significant (Table 1).

### Past obstetric and medical characteristics of study participants

The percentage of women with SMO who had history of anemia, caesarean section, and stillbirth was significantly higher than controls. Higher proportion of SMO cases had preexisting medical conditions compared to controls as well; this difference, however, did not reach statistical significance. Slightly higher percentage of women in the control group also had history of abortion compared to the SMO group, although the difference proved to be statistically insignificant (Table 2).

### Determinants of severe maternal outcome

In the bivariate binary logistic regression analysis, we assessed the association of 12 distant and proximate variables with maternal outcome. The following variables were significantly associated with SMO: parity of four or more, not attending ANC, lower gestational age, caesarean section in the current pregnancy, referral from lower level facilities, longer distance from the hospital, history of anemia, previous caesarean section and history of stillbirth. All variables considered in the bivariate analysis were significant at p = 0.25, and therefore, were eligible to be include in the multivariate binary logistic regression analysis.

In the final multivariate logistic regression model, the following variables remained independently associated with SMO: not attending ANC follow up (AOR: 4.54; CI: 3.15–6.54), caesarean section in the current pregnancy (AOR: 3.70; CI: 2.44–5.17), referral from lower level facilities (AOR: 11.71; CI: 9.01–15.23), residing more than 30 kilometers away from the hospital (AOR: 2.93; CI: 2.25–3.80), history of anemia (AOR: 2.38; CI: 1.84–3.07), and previous caesarean section (AOR: 3.33; CI: 2.06–5.38) (Table 3).

**Table 1. Distribution of socio-demographic, obstetric, prenatal and perinatal characteristics of women admitted to Keren Regional Referral Hospital.**

| Characteristics | SMOs | Controls |
|---|---|---|
| | N (%) | N (%) |
| **Age in years** | | |
| **≤ 19** | 172 (24.5) | 365 (26) |
| **20 to 34** | 317 (45.2) | 658 (47) |
| **≥ 35** | 212 (30.3) | 379 (27) |
| Parity ** | | |
| **Null** | 157 (22.4) | 347 (24.7) |
| **1 to 3** | 278 (39.7) | 629 (44.9) |
| **4 and above** | 266 (37.9) | 426 (30.4) |
| ANC attendance** | | |
| **Yes** | 533 (76) | 1323 (94.4) |
| **No** | 168 (24) | 79 (5.6) |
| Gestational age** | | |
| **< 36 weeks** | 96 (13.7) | 128 (9.1) |
| **≥ 36 weeks** | 605 (86.3) | 1274 (90.9) |
| Caesarean section in the current pregnancy** | | |
| **No** | 513 (73.2) | 1292 (92.2) |
| **Yes** | 188 (26.8) | 110 (7.8) |
| Mode of admission** | | |
| **Self-referred** | 109 (15.5) | 990 (70.6) |
| **Referred from other facilities** | 592 (84.5) | 412 (29.4) |
| Proximity of residence to hospital** | | |
| **< 30kms** | 368 (52.5) | 1173 (83.7) |
| **≥ 30Kms** | 333 (47.5) | 229 (16.3) |

Chi-square test **significant at p = 0.001

*significant at p = 0.01; *SMO* severe maternal outcome.

## Discussion

In the present study, ANC follow up, caesarean section in the current pregnancy, referral from lower level facilities, proximity of residence to hospital, history of anemia, and history of caesarean section were the major determinants of SMO.

In our study, we found that women who did not attend ANC were more likely to experience SMO. Studies from Ethiopia [7,8,14,17–20] and Brazil [9,10] also reported similar findings. There is overwhelming evidence regarding the effectiveness of ANC in preventing undesirable outcomes of pregnancy [21]. ANC provides opportunity to early detect and treat risk factors such as iron deficiency anemia and hypertensive disorders of pregnancy that can increase the odds of complication during late stage of pregnancy and delivery. Moreover, women who attend ANC tend to have better knowledge of obstetric danger signs [22] and are more likely to give birth at health facilities [23], and therefore have better chances of avoiding the occurrence of first and second delays. This suggests that a substantial number of SMO incidents can be avoided through improvements in ANC services.

Our findings further revealed that Women who underwent caesarean section in the current pregnancy had higher odds of SMO compared to women who had spontaneous vaginal delivery. This result was in line with similar studies from Brazil [24,25] that reported a three-fold increase in the risk of SMO among women who undertook caesarean section. Caesarean

**Table 2. Distribution of past obstetric and medical characteristics of women admitted to Keren Regional Referral Hospital.**

| Characteristics | SMOs | Controls |
|---|---|---|
| | N (%) | N (%) |
| **Preexisting chronic medical conditions** | | |
| **Yes** | 115 (16.4) | 187 (13.3) |
| **No** | 586 (83.6) | 1215 (86.7) |
| history of anemia ** | | |
| **Yes** | 314 (44.8) | 327 (23.3) |
| **No** | 387 (55.2) | 1075 (76.7) |
| Previous CS** | | |
| **Yes** | 81 (11.6) | 55 (3.9) |
| **No** | 620 (88.4) | 1347 (96.1) |
| history of still birth* | | |
| **Yes** | 38 (5.4) | 44 (3.1) |
| **No** | 663 (94.6) | 1358 (96.9) |
| **history of abortion** | | |
| **Yes** | 106 (15.1) | 241 (17.2) |
| **No** | 595 (84.9) | 1161 (82.8) |

Chi-square test **significant at p = 0.001

*significant at p = 0.01; *SMO* severe maternal outcome.

section is associated with several risk factors including thromboembolism, postoperative adhesion, hysterectomy, incisional hernia, and wound infections (especially if performed without administering prophylactic antibiotics) [26]. In addition, caesarean section might cause severe hemorrhage that can be threatening to the mother's life, and may require blood transfusion [27]. However, these findings should be interpreted with caution. Most of the time, women who require caesarean section are those with higher risk pregnancies or severe medical conditions. Hence, it is difficult to distinguish problems caused by the caesarean section itself from the problems caused by the conditions that require it. It has been reported that annually millions of unnecessary caesarean section are performed throughout the world [28]. Therefore, ensuring caesarean section rates stay within the recommended range (15%) [29] through promoting evidence-based practices is crucial.

This study also revealed that women who previously had caesarean section were at higher risk of developing SMO compared to women who had normal delivery in their preceding pregnancies. This finding was in agreement with previous researches from Ethiopia [8,14,19]. Studies have shown that women who had multiple caesarean sections were more likely to have problems with later pregnancies. Previous caesarean section has been recognized as a strong risk factor of uterine rupture [30]. Caesarean section leaves a scar in the uterus that weakens the elastic property of the myometrium, making the uterus easily ruptured. Moreover, the risk of placenta accrete, a potentially life threatening condition, dramatically increases with each caesarean section performed [27,31]. Previous caesarean section is also a strong risk factor for placenta previa, abruption placenta, and hysterectomy [31]. Obstetricians have different opinions on the relative advantages of vaginal birth and caesarean section following a caesarean delivery. However, in accordance with WHO recommendation, the rate of caesarean section should be restrained within the ideal range of 5–15% [29] through efforts to reduce caesarean sections performed for reasons other than medical necessity.

**Table 3. Bivariate and multivariate logistic regression analysis of factors associated with severe maternal outcome among women admitted to Keren Regional Referral Hospital.**

| Variables | COR (95% CI) | AOR (95% CI) | p-value |
|---|---|---|---|
| **Age in years** | | | |
| **< 19** | 0.98 (0.78–1.23) | 1.00 (0.73–1.36) | 0.996 |
| **21 to 34** | | 1 | |
| **≥ 35** | 1.17 (0.94–1.45) | 0.97 (0.73–1.29) | 0.847 |
| **Parity** | | | |
| **Null** | 1.02 (0.81–1.30) | 0.89 (0.65–1.22) | 0.480 |
| **1 to 3** | | 1 | |
| **4 and above** | **1.41 (1.15–1.74)** | 1.21 (0.91–1.59) | 0.190 |
| **ANC attendance** | | | |
| **Yes** | | 1 | |
| No** | **5.28 (3.97–7.02)** | **4.54 (3.15–6.54)** | **<0.001** |
| **Gestational age** | | | |
| **< 36 weeks** | **1.58 (1.20–2.10)** | 1.34 (0.94–1.93) | 0.110 |
| **≥ 36 weeks** | | 1 | |
| **Caesarean section in the current pregnancy** | | | |
| **No** | | 1 | |
| Yes** | **4.30 (3.33–5.56)** | **3.70 (2.44–5.17)** | **<0.001** |
| **Mode of admission** | | | |
| **Self-referred** | | 1 | |
| Referred from other facilities** | **13.05 (10.32–16.50)** | **11.71 (9.01–15.23)** | **<0.001** |
| **Proximity of residence to hospital** | | | |
| **< 30kms** | | 1 | |
| **≥ 30Kms** ** | **4.64 (3.78–5.69)** | **2.93 (2.25–3.80)** | **<0.001** |
| **Preexisting chronic medical conditions** | | | |
| **Yes** | 1.28 (0.99–1.64) | 1.28 (0.99–1.64) | 0.088 |
| **No** | | 1 | |
| **history of anemia** | | | |
| Yes** | **2.67 (2.20–3.24)** | **2.38 (1.84–3.07)** | **<0.001** |
| **No** | | 1 | |
| **Previous CS** | | | |
| Yes** | **3.20 (2.24–4.57)** | **3.33 (2.06–5.38)** | **<0.001** |
| **No** | | 1 | |
| **history of still birth** | | | |
| **Yes** | **1.77 (1.14–2.76)** | 1.52 (0.82–2.84) | 0.184 |
| **No** | | 1 | |
| **history of abortion** | | | |
| **Yes** | 0.86 (0.67–1.10) | 0.83 (0.60–1.14) | 0.253 |
| **No** | | 1 | |

**significant at p = 0.01

*significant at p = 0.05; *COR* Crude Odds Ratio; *AOR* Adjusted Odds Ratio; *CI* Confidence Interval.

Consistent with previous studies from Ethiopia, the current study also found that women who had history of anemia were more prone to SMO event [7,14]. Anemia increases the mother's risk to complications of pregnancy including placenta previa, abruption placenta, postpartum hemorrhage, shock, uterine sub-involution, and caesarean delivery morbidity [32]. Anemia also contributes to SMO through its negative impact on immune functions. It has

been documented that anemia increases susceptibility or severity of infections [33]. Early detection and treatment of anemia through quality and accessible antenatal care services is imperative to reduce the effect of anemia on adverse maternal outcome.

Among all factors, referral from lower-level health facilities had the strongest link to SMO in this study. Women referred from lower level facilities had more than twelve-fold higher odds of ending up with SMO compared to self-referred women. The scientific literature has documented a number of challenges in the emergency referral system that compromise the chances of survival in women referred due to obstetric complications. Distance from CEmOC facility and poor quality roads are the main challenges faced by referring facilities in transporting women with obstetric complication in most developing countries [34]. Most peripheral health facilities do not have readily available ambulance and will often have to rely on unreliable sources of transportation such as private vehicles [34–36]. Poor communication between the referring and receiving health facilities is another factor that negatively affects the effectiveness of referral system. In most instances, referring facilities fail to inform staff members of the recipient hospital in advance when referring a woman with obstetric complication [35,37]. As a result, valuable time is often lost to make necessary preparations once the woman reaches the hospital. Clinical skill of health workers in the referring facilities could also have profound effect on the efficiency of referral system [37]. Health workers may fail to recognize danger signs on time or may refer the woman without proper resuscitation and stabilization [35]. Effectiveness of the referral system can be enhanced by improving the transportation systems, raising technical skills of health workers, and establishing standard referral protocol [35,38,39].

This study also identified distance from the hospital as a risk factor for SMO. Accordingly, women residing more than 30 kilometers away from the hospital were found to be at higher odds of developing SMO. Similarly, two studies done in Ethiopia [17,20] reported that mothers who travelled 10 Kilometers or more to reach the hospital were twice as likely to develop SMO. In their classic three delays model [40], Maine and Thaddeus have clearly outlined how distance from health facility can cause both first and second delays. Long distances not only hinder reaching healthcare but also discourage women from seeking it in the first place. Women who reside in remote villages often wait until their illness gets serious before they decide to seek care, hopping that it will resolve by itself. The effect of distance is even more catastrophic when coupled with lack of transportation and poor quality roads. In addition, in most developing countries including Eritrea, specialized obstetric facilities are concentrated in the urban centers [40]. This means that women who reside in remote villages will have difficulty to reach the closest health facility, leave alone a CEmOC facility. Ensuring equitable distribution of specialist obstetric care facilities and enhancing maternity waiting home services are the most feasible solutions to mitigate the negative impact of long distance on maternal health outcome.

In the present study, we could not establish significant association between preexisting medical conditions and SMO. However, previous studies from Brazil [24] and Ethiopia [7,8,19] reported that women with preexisting medical conditions were more likely to have SMO. The proportion of women with medical comorbidities was very low in both SMO and control groups in this study. Therefore, the fact that our study was underpowered for this particular variable might be the most plausible explanation for the observed inconsistency.

## Strengths and limitations

As far as the author's knowledge is concerned, this study is the first one to assess the determinants of SMO in Eritrean setting. The sample size was large enough to give the study enough power to identify all factors associated with SMO. The sub-Saharan Africa MNM criteria was

also used instead of the WHO organ dysfunction criteria, which enabled us to identify more cases of SMO in light of the limited resources in our hospital. However, the present study had some limitations as well. Data were collected from medical registers of the hospital. Hence, potential data errors in medical records could introduce some bias. Moreover, a number of important variables including religion, ethnicity, educational level and income level were left out from this study since information on these variables is not routinely recorded in the registers. The retrospective nature of this study also prevented us from collecting data on the 'Three Delays Model' despite its importance as a determinant of SMO. Finally, even though WHO suggests a follow-up period of 42 days postpartum, we were not able to identify women who encountered SMO subsequent to hospital discharge.

## Conclusion

In this study, ANC follow up, caesarean section in the current pregnancy, referral from lower level facilities, proximity of residence to hospital, history of anemia, and history of caesarean section were found to be independently associated with SMO. These findings imply that improving access to antenatal care services will have a remarkable impact in preventing SMO. Quality ANC service is critical for early identification and correction of obstetric complications in general and iron deficiency anemia in particular. Health authorities should also be committed to reduce the rate of caesarean sections performed without sound medical indications. Moreover, improving transportation facilities, establishing robust referral protocol as well as ensuring equitable distribution of CEmOC facilities can play crucial role in reducing the incidence of SMO in Keren Hospital.

## Supporting information

**S1 Data. Raw data supporting the findings of this study.**
(SAV)

## Acknowledgments

I would like to thank the research and ethics committee of ministry of health Anseba region for granting me permission to conduct this study. I would also like to extend my gratitude to staff members of Keren Hospital who participated in the process of data collection.

## Author Contributions

**Conceptualization:** Henos Kiflom Zewde.

**Formal analysis:** Henos Kiflom Zewde.

**Investigation:** Henos Kiflom Zewde.

**Methodology:** Henos Kiflom Zewde.

**Resources:** Henos Kiflom Zewde.

**Supervision:** Henos Kiflom Zewde.

**Visualization:** Henos Kiflom Zewde.

**Writing – original draft:** Henos Kiflom Zewde.

**Writing – review & editing:** Henos Kiflom Zewde.

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
