## [Decision Letter · Decision Letter 0]

30 Mar 2023

PONE-D-22-28527Determinants of severe maternal outcome in Keren hospital, Eritrea: an unmatched case-control studyPLOS ONE

Dear Dr. Zewde HK,

Thank you for submitting your manuscript to PLOS ONE. After careful consideration, we feel that it has merit but does not fully meet PLOS ONE’s publication criteria as it currently stands. Therefore, we invite you to submit a revised version of the manuscript that addresses the points raised during the review process.

We look forward to receiving your revised manuscript.

Kind regards,

Mohan Kumar

Academic Editor

PLOS ONE

Journal Requirements:

2. For studies reporting research involving human participants, PLOS ONE requires authors to confirm that this specific study was reviewed and approved by an institutional review board (ethics committee) before the study began. Please provide the specific name of the ethics committee/IRB that approved your study, or explain why you did not seek approval in this case.

Reviewers' comments:

Reviewer's Responses to Questions

**Comments to the Author**

1. Is the manuscript technically sound, and do the data support the conclusions?

Reviewer #1: Partly

Reviewer #2: Yes

2. Has the statistical analysis been performed appropriately and rigorously? 

Reviewer #1: No

Reviewer #2: Yes

3. Have the authors made all data underlying the findings in their manuscript fully available?

Reviewer #1: Yes

Reviewer #2: Yes

4. Is the manuscript presented in an intelligible fashion and written in standard English?

Reviewer #1: Yes

Reviewer #2: Yes

5. Review Comments to the Author

Reviewer #1: Maternal adverse outcome is still becoming global health problems. Particularly in sub–Saharan Africa the problem is still unresolved. Pocket studies like this clearly indicates the extent of the problems and major contributing factors for maternal adverse outcomes. Please find below some of my comments in this article. Thank you!!

Introduction section

• Your title is about determining adverse maternal outcome whereas your introduction section more talks about MNM. it should be consistent. So, revise either of the two...

Method Section

• Nothing is described about data quality control in this study. please describe the methods you used to maintain the quality of data.

• Please describe in detail about the data collection tool whether it is valid and reliable by using statistical parameters.

• In data analysis section How you selected the variables in binary logistic regression to be candidate for Multivariate analysis??

• line 154 include the level of significancy to declare the presence of statistical association.

Result Section

• in Determinants of severe maternal outcome sub section

1. You have to report the binary logistic regression report here. total number of variables included in analysis; number of variables entered to multivariable logistic regression with chi square report.

2. In Multivariate logistic regression you have to report the AOR not OR

3. How you selected the reference group each variable? In most cases SPSS have only two options to select the reference group (either first or last). here for age in years and parity you have made the middle one as a reference group?? how??

Discussion section

• please write the result of other literature while you compare the findings

Reviewer #2: The manuscript is scientifically sound and within acceptable standards for language. As a researcher, I have the opinion that I would not have missed collecting more detailed information from the study participants which would have allowed more detailed analysis.

6. PLOS authors have the option to publish the peer review history of their article (what does this mean?). If published, this will include your full peer review and any attached files.

Reviewer #1: No

Reviewer #2: **Yes: **Dr. Gitismita Naik

---

## [Author Response · Author response to Decision Letter 0]

12 May 2023

Dear Mohan Kumar,

Thank you for giving me the opportunity to submit a revised version of the manuscript “Determinants of severe maternal outcome in Keren hospital, Eritrea: an unmatched case-control study” for publication in PLOS ONE. I appreciate the time and effort that you and the reviewers dedicated to provide feedback on my manuscript, and I am grateful for the insightful comments on and valuable improvements to my paper. I have incorporated most of the suggestions made by the reviewers. These changes are highlighted using tracked changes within the manuscript. Please see below for a point-by-point response to the editor’s and reviewers’ comments and concerns. All page and line numbers refer to the revised manuscript file with track changes. 

Thank you for considering this submission and I look forward to hearing from you.

Point-by-point response to Editor’s and Reviewer’s comments

Editor’s comments:

Please ensure that your manuscript meets PLOS ONE's style requirements, including those for file naming. The PLOS ONE style templates can be found at  https://journals.plos.org/plosone/s/file?id=wjVg/PLOSOne_formatting_sample_main_body.pdf and  https://journals.plos.org/plosone/s/file?id=ba62/PLOSOne_formatting_sample_title_authors_affiliations.pdf.

Response: I have carefully studied the PLoS ONE submission guidelines including the templates you provided me with. I have made several amendments to ensure that the manuscript conforms to the journal’s style requirements. Changes have been made in the formatting of all sections’ headings in compliance with PLoS ONE’s “body formatting guidelines”. Similarly, the captions and file names for the supporting information are now corrected in accordance with the guidelines (page 28, line 504). Moreover, the “References section” has now been moved right after the main text before the supporting information (page 22, line 383) and the “Ethical consideration” part has become part of the methods section in the revised manuscript (page 11, line 206). I believe that the manuscript now meets PLoS ONE’s style requirements. However, if there are additional style requirements that need to be addressed please let me know.

For studies reporting research involving human participants, PLOS ONE requires authors to confirm that this specific study was reviewed and approved by an institutional review board (ethics committee) before the study began. Please provide the specific name of the ethics committee/IRB that approved your study, or explain why you did not seek approval in this case. Once you have amended this/these statement(s) in the Methods section of the manuscript, please add the same text to the “Ethics Statement” field of the submission form (via “Edit Submission”).

Response: Thank you. Here, in Eritrea, there is a committee called “research review and ethics committee”. It operates at regional (provincial) and national level. All health related research proposals are first submitted to the regional ethics committee. The regional committees then decide on the ethical soundness of the proposals and they send those that are ethically acceptable to the national review and ethics committee for final approval. This paper passed through the same procedure. Accordingly I have indicated that in the “Ethical consideration” section of the manuscript (page 11, line 207-208). I have also added the same sentence in the “Ethics Statement” field of the submission form. 

Reviewer 1 comments:

Maternal adverse outcome is still becoming global health problems. Particularly in sub–Saharan Africa the problem is still unresolved. Pocket studies like this clearly indicates the extent of the problems and major contributing factors for maternal adverse outcomes. Please find below some of my comments in this article. Thank you!!

Response: Thank you for your positive evaluation of the manuscript.

Introduction section: Your title is about determining adverse maternal outcome whereas your introduction section more talks about MNM. it should be consistent. So, revise either of the two...

Response: Thank you for pointing this out. The reviewer is right. Even though the focus of this study is severe maternal outcome (SMO), the introduction section of the original manuscript put undue focus on maternal near miss (MNM). In response to this comment, I have made major revision to the introduction section of the manuscript. I have removed the third paragraph of the original manuscript in its entirety as it contains unnecessary details that the readers can easily access from other sources (page 5, line 84). In addition, I have added a new paragraph that focuses on the importance of maternal death review and its shortcomings (page 3, line 56). Some minor changes were also applied throughout the introduction section to make it fit with the study context and to stress on the importance of the study (lines 67-68 & lines 95-105). 

Method Section: Nothing is described about data quality control in this study. please describe the methods you used to maintain the quality of data. 

Response: This is a valuable comment from reviewer 1. Data quality control should be an integral part of any research article. The original manuscript states nothing on the methods we used to ensure data quality. We took several measures to make sure that our data is of high quality even though we failed to mention it in the original manuscript. In the revised manuscript we have incorporated a new sub-section under the heading “Data quality assurance” to describe all the measures we took to ensure data quality (page 10, line 194).

Method Section: Please describe in detail about the data collection tool whether it is valid and reliable by using statistical parameters.

Response: Thank you for your remark. In compliance with this comment, we have rephrased and rewritten the “Data collection procedure and materials” sub-section of the “Methods section”. We have provided finer details regarding the data abstraction tool we used for the purpose of this study and the methods we employed to ensure its validity and reliability (page 7, lines 138-152).

Method Section: In data analysis section How you selected the variables in binary logistic regression to be candidate for Multivariate analysis??

Responses: Thank you for pointing this out. Again, this is another important piece of information that was missed out from the original manuscript. We used p=0.25 in the bivariate analysis as a cut-off point to include variables in the multivariate logistic regression analysis. We have included a sentence stating this in the revised version of the manuscript (page 9, lines 185-186).

Method Section: line 154 include the level of significancy to declare the presence of statistical association.

Responses: Again, thank you. The level of significance we used for both bivariate and multivariate logistic regression analysis is p=0.05. We have included a sentence in the “Statistical analysis” part of the manuscript to indicate this (page 9, lines 188-189).

in Determinants of severe maternal outcome sub section: You have to report the binary logistic regression report here. total number of variables included in analysis; number of variables entered to multivariable logistic regression with chi square report.

Responses: In light of the suggestion made by reviewer 1, we have added additional paragraph in the “Determinants of severe maternal outcome” sub-section to incorporate all the points mentioned in his/her comments (pages 14-15, lines 249-255). We have also added additional column in “Table 3” to include the crude odds ratio (COR) obtained from the binary logistic regression analysis along with its 95% confidence interval. A column was also included to report the p-value for the final model of multivariate logistic regression analysis (page 15, line 262). 

in Determinants of severe maternal outcome sub section: In Multivariate logistic regression you have to report the AOR not OR

Responses: Thank you for spotting this problem. Indeed, in the multivariate logistic regression report AOR should be reported instead of OR. It was written OR by mistake and therefore we have replaced the “OR” by “AOR” in all incidents throughout the paragraph (page 15, lines 257-261). The same correction has also been made in the “Results” sub-section of the “Abstract” section in the revised manuscript.

in Determinants of severe maternal outcome sub section: How you selected the reference group each variable? In most cases SPSS have only two options to select the reference group (either first or last). here for age in years and parity you have made the middle one as a reference group?? how??

Responses: this is a good remark from reviewer 1. In all variables included in this study we used the first variable category as a baseline (reference) category. Hence, we coded all variables in a way that suits this method. All variable categories that we assumed have the least odds of encountering SMO were coded as first. The reason behind this decision is that we want to increase the chances of our odds ratio being greater than 1 as such result is easy to interpret compared to odds ratio result of less than 1. Consequently, the coding for the variable categories in SPSS for most variables (especially for those that have more than two categories) is not necessarily the same as the logical order of the variable categories reported in the tables within the manuscript. If we take the variable “age” as an example, age group “20 to 34” is placed in the middle in the reported tables. In SPSS however, this age category are code as first in keeping with our assumption that this age group have the lowest odds of SMO compared to the remaining two age groups. If the reviewer has access for the supplementary material (row data supporting the results of this manuscript) he/she can check it for him/herself.

Discussion: please write the result of other literature while you compare the findings

Responses: I am not sure if I correctly understood this comment given by reviewer 1. I think the reviewer is suggesting that we include the exact findings of the literatures we cited in our discussion section when comparing to our own findings. In most cases we included two or more literatures for each significant finding in our study as a comparison. Hence, it is difficult to mention the findings from the comparison studies in detail. In view of that, we did not include any details (such as exact odds ratio) in most cases. However we tried to indicate the strength of the association in some instances where we feel it would be appropriate. For example, “…. twice as likely to develop SMO” (page 20, line 339) and “…. similar studies from Brazil [29, 30] that reported a three-fold increase in the risk of SMO …..” (page 18, line 282).

Reviewer 2 comments: 

The manuscript is scientifically sound and within acceptable standards for language. As a researcher, I have the opinion that I would not have missed collecting more detailed information from the study participants which would have allowed more detailed analysis.

Response: Thank you for your positive evaluation of the manuscript. I agree with the reviewer’s assessment. Our study would have benefited tremendously had it included more variables that are potentially associated with SMO. However, this study was based on secondary data collected from medical registers and patient cards of the hospital which limited our ability to collect information on more variables than those included in the manuscript. We have discussed this limitation in the “strengths and limitations” sub-section of the “Discussion” section in detail (page 21, line 266-269).

---

## [Editor Report · Decision Letter 1]

12 Sep 2023

PONE-D-22-28527R1Determinants of severe maternal outcome in Keren hospital, Eritrea: an unmatched case-control studyPLOS ONE

Dear Dr. Zewde,

Thank you for submitting your manuscript to PLOS ONE. After careful consideration, we feel that it has merit but does not fully meet PLOS ONE’s publication criteria as it currently stands. Therefore, we invite you to submit a revised version of the manuscript that addresses the points raised during the review process. The introduction speaks to maternal near miss and its association with SMO but the study is not about MNM. This point has been previously pointed out. Please align the introduction with the research question.It would also help if you included the SMOs and comorbidities for the cases and controls. Minor editing is needed to ensure that the writing is consistent with academic writing i.e.1. Use the same fonts throughout the manuscript.2. Capitalise words as appropriate i.e. proper nouns and when starting a sentence3. Don't start a sentence with a number. 4. The use of bold text in the manuscript Please submit your revised manuscript by Oct 27 2023 11:59PM with tracked changes and a clean manuscript.  If you will need more time than this to complete your revisions, please reply to this message or contact the journal office at plosone@plos.org. Please include the following items when submitting your revised manuscript:A rebuttal letter that responds to each point raised by the academic editor and reviewer(s). You should upload this letter as a separate file labeled 'Response to Reviewers'.A marked-up copy of your manuscript that highlights changes made to the original version. You should upload this as a separate file labeled 'Revised Manuscript with Track Changes'.An unmarked version of your revised paper without tracked changes. You should upload this as a separate file labeled 'Manuscript'.If applicable, we recommend that you deposit your laboratory protocols in protocols.io to enhance the reproducibility of your results. Protocols.io assigns your protocol its own identifier (DOI) so that it can be cited independently in the future. For instructions see: https://journals.plos.org/plosone/s/submission-guidelines#loc-laboratory-protocols. Additionally, PLOS ONE offers an option for publishing peer-reviewed Lab Protocol articles, which describe protocols hosted on protocols.io. Read more information on sharing protocols at https://plos.org/protocols?utm_medium=editorial-email&utm_source=authorletters&utm_campaign=protocols.

We look forward to receiving your revised manuscript.

Kind regards,

Mergan Naidoo, PhD

Academic Editor

PLOS ONE
---

## [Author Response · Author response to Decision Letter 1]

20 Oct 2023

Dear Dr. Mergan Naidoo,

Thank you for giving me the opportunity to submit a revised version of the manuscript “Determinants of severe maternal outcome in Keren hospital, Eritrea: an unmatched case-control study” (Manuscript ID: PONE-D-22-28527R1) for publication in PLOS ONE. I have made necessary changes to the manuscript according to the suggestions provided during the review process. These changes are highlighted using tracked changes within the manuscript. Please see below for a point-by-point response to the comments. All page and line numbers refer to the revised manuscript file with track changes. 

Thank you for considering this submission and I look forward to hearing from you.

Point-by-point response to Editor’s comments

Response to Editor’s comments:

1. The introduction speaks to maternal near miss and its association with SMO but the study is not about MNM. This point has been previously pointed out. Please align the introduction with the research question.

Response: Thank you for reminding me that the introduction section needs further amendments. I have rewritten major portion of the introduction section in the revised version of the manuscript. Now, the introduction mainly discusses the background on the determinants of maternal mortality and morbidity and on the aim of the study in relation to the context of previous studies. I have also made similar amendments in the “Background” sub-section of the “Abstract” section. I believe that the introduction section is now in tandem with the title and content of the man uscript. However, if there are additional issues that need to be addressed please let me know.

2. It would also help if you included the SMOs and comorbidities for the cases and controls.

Response: In this study, women who experienced severe maternal outcome (SMO) were considered as cases (i.e. women who encountered one or more organ dysfunction and women who died). Women who delivered without serious obstetric complications, on the other hand, were enrolled as controls. Hence, only women with SMO (cases) can be categorized as maternal deaths and maternal near misses. I have indicated in the results section that there were 662 near misses and 39 maternal deaths.

3. Use the same fonts throughout the manuscript.

Response: Thank you. In light of this comment, I have carefully assessed the entire manuscript for consistency of fonts. The font, “Times New Roman”, is used consistently throughout the manuscript. Moreover, font size “12” is applied for the entire text, except for the headings which are formatted in accordance with the PLoS ONE’s “Manuscript body formatting guidelines”.

4. Capitalise words as appropriate i.e. proper nouns and when starting a sentence

Response: Thank you for this comment. There were some instances where capitalization was not properly use in the former manuscript. Necessary corrections have now been made in the revised manuscript.

5. Don't start a sentence with a number.

Response: After careful revision, I have paraphrased all such sentences to avoid starting a sentence with a number.

6. The use of bold text in the manuscript

Response: again, thank you. In accordance with this comment, the revised manuscript does not contain any bold text with in the main body except for headings.

Response to journal requirements:

Response: I have checked each and every reference in the manuscript, and I assure you that there is no any retracted paper cited in the reference list. I have made a number of changes to the reference list based on the comments given in the review process. I have indicated all changes made to the reference list in the revised manuscript file with track changes (including references removed from and added to the revised manuscript). In addition, I have added the doi for all peer-reviewed articles in the reference list. I have also added a link for the remaining references along with the date when they were accessed.

---

## [Editor Report · Decision Letter 2]

27 Dec 2023

PONE-D-22-28527R2Determinants of severe maternal outcome in Keren hospital, Eritrea: an unmatched case-control studyPLOS ONE

Dear Dr. Zewde,

Thank you for submitting your revised manuscript to PLOS ONE. After careful consideration, we feel that it has merit but does not fully meet PLOS ONE’s publication criteria as it currently stands. Therefore, we invite you to submit a revised version of the manuscript that addresses the points raised during the review process. In the abstract your conclusion must be aligned to your aim.Please improve the quality of the academic writing by editing the manuscript. Some of the issues identified include: Not capitalizing proper nouns, using a number to start a sentence and the grammar.Please ensure that the references comply with the Vancouver referencing format especially when using online databases and websites.

Please submit your revised manuscript by Feb 10 2024 11:59PM. If you will need more time than this to complete your revisions, please reply to this message or contact the journal office at plosone@plos.org. Please include the following items when submitting your revised manuscript:A rebuttal letter that responds to each point raised by the academic editor and reviewer(s). You should upload this letter as a separate file labeled 'Response to Reviewers'.A marked-up copy of your manuscript that highlights changes made to the original version. You should upload this as a separate file labeled 'Revised Manuscript with Track Changes'.An unmarked version of your revised paper without tracked changes. You should upload this as a separate file labeled 'Manuscript'.If applicable, we recommend that you deposit your laboratory protocols in protocols.io to enhance the reproducibility of your results. Protocols.io assigns your protocol its own identifier (DOI) so that it can be cited independently in the future. For instructions see: https://journals.plos.org/plosone/s/submission-guidelines#loc-laboratory-protocols. Additionally, PLOS ONE offers an option for publishing peer-reviewed Lab Protocol articles, which describe protocols hosted on protocols.io. Read more information on sharing protocols at https://plos.org/protocols?utm_medium=editorial-email&utm_source=authorletters&utm_campaign=protocols.

We look forward to receiving your revised manuscript.

Kind regards,

Mergan Naidoo, PhD

Academic Editor

PLOS ONE
---

## [Author Response · Author response to Decision Letter 2]

27 Jan 2024

Dear Dr. Mergan Naidoo,

Thank you for giving me the opportunity to submit a revised version of the manuscript “Determinants of severe maternal outcome in Keren hospital, Eritrea: an unmatched case-control study” (Manuscript ID: PONE-D-22-28527R2) for publication in PLOS ONE. I have made necessary changes to the manuscript according to the suggestions provided during the review process. These changes are highlighted using tracked changes within the manuscript. Please see below for a point-by-point response to the comments. All page and line numbers refer to the revised manuscript file with track changes. 

Thank you for considering this submission and I look forward to hearing from you.

Point-by-point response to Editor’s comments

Response to Editor’s comments:

1. In the abstract your conclusion must be aligned to your aim.

Response: In accordance with the editor’s comments, I have rewritten the conclusion part of the abstract section. Now, the conclusion focuses on the main question the manuscript tries to address and the key recommendations for improvement.

2. Please improve the quality of the academic writing by editing the manuscript. Some of the issues identified include: Not capitalizing proper nouns, using a number to start a sentence and the grammar.

Response: Thank you for this comment. There were some instances where capitalization was not properly used in the former manuscript. Necessary corrections have now been made in the revised manuscript. I have also carefully revised the manuscript to make sure that there are no sentences that start with a number. Moreover, all necessary amendments were made to avoid grammatical mistakes in the revised manuscript. If you still think that there are unresolved issues in this area, please let me know. It would also be helpful if you could mention one or two examples where such errors remain uncorrected.

3. Please ensure that the references comply with the Vancouver referencing format especially when using online databases and websites.

Response: I have carefully studied the guide for the Vancouver referencing style and I have made minor adjustments to the reference list to make sure that it adheres with the recommended format.

Response to journal requirements:

Response: I have checked each and every reference in the manuscript, and I assure you that there is no any retracted paper cited in the reference list. I have indicated in this rebuttal letter all the changes made to the reference list.

---

## [Editor Report · Decision Letter 3]

14 Feb 2024

Determinants of severe maternal outcome in Keren hospital, Eritrea: an unmatched case-control study

PONE-D-22-28527R3

Dear Dr. Henos Kiflom Zewde

We’re pleased to inform you that your manuscript has been judged scientifically suitable for publication and will be formally accepted for publication once it meets all outstanding technical requirements.

Kind regards,

Mergan Naidoo, PhD

Academic Editor

PLOS ONE
---

## [Editor Report · Acceptance letter]

30 Apr 2024

PONE-D-22-28527R3 

PLOS ONE

Dear Dr. Zewde, 

I'm pleased to inform you that your manuscript has been deemed suitable for publication in PLOS ONE. Congratulations! Your manuscript is now being handed over to our production team.

Kind regards, 

on behalf of

Professor Mergan Naidoo 

Academic Editor

PLOS ONE